# Immune Thrombocytopenia in Antiphospholipid Syndrome: Is It Primary or Secondary?

**DOI:** 10.3390/biomedicines9091170

**Published:** 2021-09-06

**Authors:** Riccardo Tomasello, Giulio Giordano, Francesco Romano, Federica Vaccarino, Sergio Siragusa, Alessandro Lucchesi, Mariasanta Napolitano

**Affiliations:** 1Department of Health Promotion, Mother and Child Care, Internal Medicine and Medical Specialties (ProMISE), University of Palermo, 90141 Palermo, Italy; riccardot1993acp@gmail.com (R.T.); francescoromanomd@gmail.com (F.R.); fedevaccarino@gmail.com (F.V.); sergio.siragusa@unipa.it (S.S.); 2Division of Internal Medicine, Hematology Service, Regional Hospital “A. Cardarelli”, 86100 Campobasso, Italy; giuliogiordano@hotmail.com; 3Haematology Unit, IRCCS Istituto Romagnolo per lo Studio dei Tumori “Dino Amadori” (IRST), 47104 Meldola, Italy; ale.lucchesi@gmail.com

**Keywords:** antiphospholipid antibodies, antiphospholipid syndrome, thrombocytopenia, lupus anticoagulant, immune thrombocytopenia

## Abstract

Antiphospholipid syndrome (APS) is frequently associated with thrombocytopenia, in most cases mild and in the absence of major bleedings. In some patients with a confirmed APS diagnosis, secondary immune thrombocytopenia (ITP) may lead to severe thrombocytopenia with consequent major bleeding. At the same time, the presence of antiphospholipid antibodies (aPL) in patients with a diagnosis of primary ITP has been reported in several studies, although with some specific characteristics especially related to the variety of antigenic targets. Even though it does not enter the APS defining criteria, thrombocytopenia should be regarded as a warning sign of a “high risk” APS and thus thoroughly evaluated. The presence of aPL in patients with ITP should be assessed as well to stratify the risk of paradoxical thrombosis. In detail, besides the high hemorrhagic risk in secondary thrombocytopenia, patients with a co-diagnosis of APS or only antibodies are also at risk of arterial and venous thrombosis. In this narrative review, we discuss the correlation between APS and ITP, the mechanisms behind the above-reported entities, in order to support clinicians to define the most appropriate treatment strategy in these patients, especially when anticoagulant or antiplatelet agents may be needed.

## 1. Introduction

Antiphospholipid antibody syndrome (APS) is an autoimmune acquired thrombophilia characterized by recurrent thrombosis and pregnancy morbidity in the presence of antiphospholipid antibodies (aPL), namely anticardiolipin antibodies (aCL), anti-β2 glycoprotein I (Anti-β2GPI) antibodies, and lupus anticoagulant (LA) [1,2,3]. The aPL are a heterogenous group of antibodies that react with phospholipids (PLs), PL-protein complexes, and PL-binding proteins. The main antigenic targets of these antibodies are β2GPI and prothrombin, which together account for more than 90% of the antibody-binding activity in APS; antigenic targets also include tissue plasminogen activator (tPA), phosphatidylserine (PS), plasmin, annexin 2, activated protein C (APC), thrombin, antithrombin III (AT-III), and annexin V [4,5,6,7,8,9,10]. Antiphospholipid antibodies represent the strongest acquired risk factors for arterial and venous thrombosis and also the most common acquired thrombophilia. Clinical symptoms of APS include thrombosis in any blood vessel of any organ, with no substantial differences between veins and arteries.

Primary immune thrombocytopenia (ITP) is an autoimmune disorder characterized by isolated thrombocytopenia due to both immune-mediated platelet destruction by antiplatelet antibodies that generally recognize platelet membrane glycoproteins (GPs) and impaired platelet production without other causes or disorders that may be associated with low platelets count. Primary ITP leads to an increased risk of bleeding, although some patients may be asymptomatic [11].

### 1.1. Diagnosis of APS

Diagnosis of APS is currently based on the Sapporo criteria, updated in 2006 [12], Table 1. Diagnosis is based on the fulfillment of at least one clinical and one laboratory criteria.

Available cohort studies have evaluated both the frequency of thrombocytopenia in patients with APS and the frequency of aPL in patients affected by thrombocytopenia; thrombocytopenia is currently considered as a “non criteria” manifestation for APS diagnosis; this is, however, still debated. The last reported meeting of the Technical Task Force Report on Antiphospholipid Syndrome Clinical Features, suggested, in fact, to consider thrombocytopenia as a classification criterion of APS [13].

Prospective multicenter studies aiming to better define also the characteristics and the role of thrombocytopenia in APS are currently ongoing [14] with the aim of improving the management of APS and identifying patients at higher risk of thrombosis.

### 1.2. Diagnosis of Primary ITP

Diagnosis of primary immune thrombocytopenia (ITP) is considered one of exclusion and requires an accurate differential diagnosis with secondary forms as in these cases, thrombocytopenia may be responsive only to the treatment of the underlying disease. Furthermore, the administration of treatment for primary ITP in some cases of secondary ITP could even worsen the disease (e.g., APS-associated ITP and unilinear myelodysplastic syndrome). The patient interview should focus on the exposure to any of the several potential triggers of ITP (drugs, vaccines, and transfusions) as well as on the evaluation of risk factors for secondary forms (weight loss, autoimmune diseases, chronic infections—mainly viral, such as HIV, HBV, or HIV). Family history is important to rule out cases of inherited thrombocytopenia that may be related to genetically determined platelet disorders. During physical assessment, the clinician must search for potential sites of bleeding (cutaneous and mucosal) and identify suggestive signs of secondary ITP (hepatosplenomegaly, abdominal masses, lymphadenopathies, and bone pain), [11,15].

Peripheral blood smear examination is also mandatory to exclude a pseudothrom-bocytopenia or platelet satellitism.

### 1.3. The Coexistence of APS and ITP

Harris et al. first reported the presence of aPL within the course of ITP [16], while Bidot et al. linked the levels of antiphospholipid antibodies with the exacerbations and remissions of ITP [17]. In the same article, Bidot reports some interesting findings related to differences between antibodies characteristics in ITP and APS:IgG and IgM antibodies against b2GP1 and IgM antibodies to FVII/VIIa are more common in APS, as well as IgM against phospholipids (aCL, aPC, aPS, aPE);IgG antibodies against phospholipids are more common in ITP and generally recognize fewer antigens (<3) than the antibodies detected in APS (>3).

This particular finding could be explained by the greater variety of target tissues in APS, while in ITP, antigen targets are merely platelet membrane glycoproteins.

The issue of ITP during APS as a primary or secondary condition thus arises. Interestingly, as stated by Stasi et al., the administration of immunosuppressive therapies to treat thrombocytopenia during APS reduces the titers of anti-GP antibodies that could be detected in APS (and in ITP without APS as well), but not the titers of aPL [18], thus suggesting that thrombocytopenia during APS could be a clinically independent entity due to antibodies different from those detected in ITP. One study by Galli et al. showed that about 40% of patients with thrombocytopenia and APS had antibodies directed against the GPIIb-IIIa or GPIb-IX-V complexes [19].

## 2. APS and Thrombocytopenia

Thrombocytopenia occurs in APS with a frequency ranging from 20% to 50%, and the estimated bleeding risk associated with it is much lower than the thrombotic risk associated with aPL. That said, it is important to explore the mechanism behind thrombocytopenia in order to establish the most appropriate treatment strategy in these patients, especially when anticoagulant or antiplatelet agents are needed [12].

The pathogenesis of thrombocytopenia in antiphospholipid antibody-positive patients is not fully understood. However, based on currently available data [20,21,22,23,24,25,26], we here summarize four different mechanisms, reported in Table 2.

### Clinical Significance of APL-Positivity in ITP

An international consensus introduced the definition of “aPL-associated thrombocytopenia” when the laboratory findings of aPL are associated with thrombocytopenia in the absence of APS-defining criteria [3].

According to ITP guidelines, when aPL are detected in an ITP patient without a history of thrombosis or obstetric complications, this finding will not change the diagnosis of primary ITP nor the recommended treatment (with an important caveat on thrombopoietin receptor agonists, discussed in a dedicated section). The latest ASH “ITP Practice and Guideline Panel” on the contrary, stated that evaluating ITP patients for aPL is unnecessary [27] due to the lack of good evidence of clinical association [28]. Several studies, on the other hand, go in a different direction:A study by Diz-Kücükkaya et al. found that 45.1% of a cohort of ITP patients who were persistently positive for aPL developed later APS [29];A study by Machin et al. showed that the statistically most relevant risk factor for thrombosis in patients with thrombocytopenia is the co-diagnosis of APS, over an average 5-years follow-up [30];A study by Hisada et al. found that the combination of aPL and thrombocytopenia doubled the risk of future thrombosis over an average 10-years follow-up [27]A study by Funauchi et al. demonstrated that women with ITP and aPL-positivity had increased thrombosis and obstetric complications risks when compared to the aPL-negative group [31];An interesting cross-sectional study reported that the platelet counts of patients with a diagnosis of high-risk APS (triple positive: LAC, anti-beta2-GPI, and aCL) decreased earlier before the appearance of a full clinical picture of CAPS [32]. Therefore, the screening for aPL may identify a subgroup of ITP patients at higher risk of thrombosis.The already mentioned International Consensus Panel stated that thrombocytopenia occurring in patients with persistent aPL-positivity is associated with increased thrombotic risk and therefore should be considered different from simple ITP [3];Data from the Italian Registry of Antiphospholipid Antibodies reported that 40% of the APS patients with moderate thrombocytopenia and 9% of the APS patients with severe thrombocytopenia developed thrombosis [24];A review by Frison et al. on the records of 233 outpatients with primary or secondary thrombocytopenia (platelet count < 100 × 109/L) concluded that triple-positive patients had a significantly lower median platelet count compared to other patients with aPL-positivity [33].

Even though it does not enter the APS defining criteria, thrombocytopenia should be regarded as a warning sign, at least in the assessment of high-risk APS and thoroughly evaluated. At the same time, the presence of aPL during ITP should be assessed to stratify the risk of thrombosis.

## 3. Clinical Significance of Thrombocytopenia and aPL-Positivity in Patients with Systemic Lupus Erythematosus

Thrombocytopenia can be detected in patients with systemic lupus erythematosus (SLE) with a frequency that ranges from 20% to 40%. It is usually mild (platelets count >50 × 10^9^/L), but it can also occur as severe (platelets count <25 × 10^9^/L), thus requiring specific treatment. In a few cases, thrombocytopenia can be observed as an adverse event secondary to immunosuppressive treatments, such azathioprine, methotrexate, and more rarely, hydroxychloroquine (HCQ). In these particular cases, the pathogenesis is drug-related and can involve other cytopenias [34].

The majority of autoantibodies found in SLE-associated thrombocytopenia are antiplatelet surface glycoproteins (i.e., anti-GPIIb/IIIa), similar to those found in the course of ITP, but the exposure of platelet to cardiolipin-like membrane phospholipids can also lead to the production of anticardiolipin antibodies. A study by Nakamura et al. explored the role of anti-CD40L antibodies that can mediate the interaction between CD40L of T-cell surface and CD40 antigen on B-cell surface, thus promoting B cell activation and antibodies production [35]. As observed in ITP, these antibodies may not be specific; they can be detected in 30% to 70% of cases. However, their titers can be reduced by immunosuppressive therapy while treatment discontinuation induces their increase, thus indicating a role in the pathogenesis of SLE-associated thrombocytopenia [36].

Antiphospholipid antibodies can be found in patients with SLE with a frequency that ranges from 30% to 40%. They are associated with a higher prevalence of complications of SLE, such as venous and arterial thrombosis, pregnancy morbidity, secondary immune thrombocytopenia, hemolytic anemia, renal microangiopathy, cardiovascular and cerebrovascular diseases. The most common aPL found in the course of SLE are LA and aCL [37].

The incidence of secondary immune thrombocytopenia is known to be higher in SLE patients with high titers of aCL or LA, especially for strong LA positivity. A review by Love et al. and a meta-analysis by Chock et al. demonstrated that the risk of thrombocytopenia in SLE patients with strong LA positivity was at least double than LA-negative subjects [38,39].

### 3.1. Therapeutic Management of SLE-Associated Thrombocytopenia

Thrombocytopenia in SLE is rarely severe, and most patients have a platelet (PLT) count around 50 × 10/L without bleeding manifestations. However, in some cases, it may require emergency treatment to prevent hemorrhagic complications. This is commonly performed with high-dose glucocorticoids, with or without high-dose intravenous immunoglobulins. Usually, this treatment allows achieving a complete platelet response; in case of treatment failure, several immunosuppressive agents, such as HCQ, danazol, azathioprine, cyclosporine, mycophenolate mofetil, cyclophosphamide, and anti-CD20 monoclonal antibodies (rituximab), can be implemented in the second line. In the case of recurrent/refractory thrombocytopenia, a splenectomy can be performed to achieve a durable response. Thrombopoietin receptor agonists (TPO-RAs), Eltrombopag and Romiplostim, were approved in 2008 as a second line therapy for refractory ITP and few studies have reported their efficacy in immune cytopenias, such as SLE-associated thrombocytopenia. Some reports have shown a complete remission of thrombocytopenia achieved by single-agent TPO-RA treatment [36].

An algorithm for first- and second-line treatment of SLE-associated thrombocytopenia is reported in Figure 1.

Novel steroid-sparing therapies include rituximab, administered at various dose regimens ranging from 100 to 200 mg every week for 1–4 weeks and 1000 mg on a single dose. Response to this last therapy is highly variable and generally not prolonged [36].

### 3.2. Prevention of aPL-Associated Complications in SLM

Antiphospholipid antibodies represent the strongest acquired risk factors for arterial and venous thrombosis. It is thus strongly recommended to monitor other risk factors patients may have [37].

High-risk aPL profiles without a history of thrombosis, such as triple-positive asymptomatic carriers, are treated in prophylaxis with low-dose acetylsalicylic acid (LDA). However, this practice is not supported by evidences from randomized controlled trials [37].

The current recommendations on the management of SLE suggest treating high-risk aPL profiles similarly to primary APS, given the increased risk of aPL-related morbidities in these patients [37]. Anticoagulant treatment should be carefully administered in patients with SLE and persistent aPL-positivity who experience venous or arterial thrombosis and have concomitant thrombocytopenia. In the case of moderate-severe thrombocytopenia (Platelet count < 50× 10^9^/L), any treatment with vitamin K-antagonists should be discontinued, and low molecular weight heparin or fondaparinux should be given until platelet count recovery. The treatment recommendations for APS and persistent aPL-positivity are summarized in Table 3.

## 4. An Overview of the Treatment of Antiphospholipid Antibodies Syndrome: The Latest Guidelines

The reported risk of thrombosis in aPL-positive patients with no prior thrombotic episodes or other risk factors (e.g., autoimmune disease) is <1% per year. Thus, the administration of LDA for “primary prevention of thrombosis” in patients with APS but no prior thrombosis (asymptomatic aPL carriers) is controversial [40]. According to a recent meta-analysis, LDA has been associated with significant risk reduction in arterial but not in venous thrombosis when compared to placebo [41].

The protective role of LDA for the primary prophylaxis of thrombosis is, up to date, not supported by robust evidences, also for those patients with high-risk persistent aPL positivity. Primary prevention of thrombosis should be thus defined on an individual basis in patients with high-risk aPL. It should be based on regular assay of antibodies titer and control of any other modifiable risk factor for thrombosis, along with the treatment of any underlying known autoimmune disease [42], whether or not within this context. Risk stratification, which may also include additional “non criteria” manifestations, such as thrombocytopenia, needs to be clarified yet [43,44].

Standard care for thrombotic APS is long-term (LT) anticoagulation with a vitamin K antagonist (VKA) [45]. There is currently insufficient evidence to recommend the use of direct oral anticoagulants (DOACs) over VKA in thrombotic APS. The largest studies conducted on this topic showed that DOACs were associated with an increase in hemorrhagic adverse events and an increased thrombosis incidence when compared to VKA [46].

While the recommendation on VKA in APS with a history of venous thromboembolism (VTE) is strong, it is not yet clear if patients with APS and arterial thromboses can really benefit from long-term anticoagulation with VKA rather than prophylaxis with antiplatelet agents, such as LDA. Many experts, in fact, recommend VKA treatment for patients with APS and a history of arterial thromboses because they have a general tendency to recur, thus suggesting that a therapeutic approach based on long-term anticoagulation may be safer than LDA alone. Some authors indicate maintaining a INR target of 3–4 for APS patients with recurrent arterial thrombosis (versus 2–3 of APS patients with a history of recurrent VTE) [34]; however, patients with a history of stroke and a low-titer of aCL may be treated with LDA alone [45].

Pregnant women with thrombotic APS should be treated with LDA in association with therapeutic-dose heparin, regardless of the pregnancy history. LDA or prophylactic-dose low molecular weight heparin (LMWH) can be used in pregnant women with APS but no prior history of thrombosis (including 6 weeks postpartum) [45].

A combined therapy of anticoagulant treatment, glucocorticoids, plasma exchange (PEX), and intravenous high dose immunoglobulin (HD-IVIG) can be used in catastrophic APS (CAPS). Cyclophosphamide can also be used in CAPS when associated with a secondary autoimmune disease, such as SLE. Rituximab can be used in refractory cases, after failure or inability to take the above-mentioned combined therapies or in the presence of micro-angiopathic hemolytic anemia [45], Table 3.

## 5. Management of aPL-Associated Thrombocytopenia

APS-associated thrombocytopenia is usually mild (70–120 × 10^9^/L) and benign. In the majority of cases, no intervention is required. However, in a few cases, thrombocytopenia may be severe (platelet count < 25 × 10^9^/L, WHO grade 4) and require aggressive treatment. Generally, APS patients with thrombocytopenia can be given platelet aggregation inhibitors (i.e., LDA) and/or anticoagulant therapy (i.e., warfarin). Anti-thrombotic treatment should be stopped only in case of severe thrombocytopenia or bleeding [45].

aPL-positive patients with thrombocytopenia without the clinical manifestations of APS are diagnosed and treated as primary ITP. For the same reason, APS-associated severe thrombocytopenia is treated as primary ITP (with the notable exception of thrombopoietin receptor agonists) [47].

Glucocorticoids (i.e., prednisone 1 to 2 mg/kg/day, orally) and high-dose immunoglobulins (IVIG) are used in the first-line treatment of APS-associated thrombocytopenia and aPL-positive ITP. Second-line treatment for those being unresponsive or intolerant to glucocorticoids, can include immunosuppressive drugs, such as azathioprine and cyclophosphamide [47].

### 5.1. Definition of Clinical Response

The definition of response to the treatment of immune thrombocytopenia has been reported by Rodeghiero et al. [11]: complete Response (CR) is defined by PLT count ≥ 100 × 10^9^/L, and Response (R) indicates PLT count ≥ 30 × 10^9^/L and at least twice the initial value. Failure to respond to splenectomy is included in the definition of “refractory” and according to Rodeghiero et al., patients whose platelet counts do not respond to two treatments or more can also be considered as “refractory”. Some evidences suggest not to be too fast in declaring the absence of response to treatment [48], in particular when immunomodulatory drugs are administered. Thus, the timing for evaluation of response to the treatment of thrombocytopenia is related to the type of treatment adopted. In the front-line therapy of immune thrombocytopenia, recent guidelines recommend limiting the administration of corticosteroids (including full dose and tapering) to not more than eight weeks for their potential side effects [49].

For steroids and immunoglobulins, a time frame of one week could be appropriate for the initial evaluation of response to treatment [50]. For other available second-line therapies, such as rituximab and hydroxychloroquine, the time to reach an adequate response may be longer, up to four/six weeks.

### 5.2. The Role of anti-CD20 MoAb in Treating aPL-Associated Thrombocytopenia

The chimeric anti-CD20 monoclonal antibody rituximab is commonly adopted as a second-line treatment for immune thrombocytopenia or other immune cytopenias; available studies have reported the efficacy of this treatment with response rates at 1 year ranging from 50% to 60% for immune thrombocytopenia [51] to 75% for warm autoimmune hemolytic anemias [52]. Rituximab has also been evaluated for its efficacy on renal and extra-renal symptoms of SLE. However, primary study endpoints were not reached in two trials [53,54]. An observational cohort study reported in patients with SLE treated with ituximab [55] a clinical response in up to 70% of cases. A pilot phase 2 study confirmed the safety of rituximab in aPL-positive patients and suggested, even in the absence of any change in aPL levels, the efficacy of rituximab in controlling some non criteria manifestations of APS [56]. There are no controlled clinical trials exploring if rituximab is effective in the antiphospholipid syndrome. However, it could balance the effect of bleeding and thrombosis in APS patients with severe thrombocytopenia. By reducing the production of autoantibodies, rituximab could simultaneously raise the platelets count and reduce the risk of thrombosis [40].

### 5.3. Thrombopoietin Receptor Agonists in the II Line Treatment of Connettive Tissue Disease-Associated Thrombocytopenia

Thrombopoietin receptor agonists are a novel class of molecules that interact with the thrombopoietin receptor exposed by hematopoietic stem cells promoting proliferation and differentiation of megakaryocytes in the bone marrow, thus increasing peripheral blood functioning platelet count through a dose-dependent mechanism [57].

The TPO-RA Eltrombopag was approved by FDA in 2008 for the treatment of chronic ITP with a solid efficacy and safety profile that has been well defined in the course of the last ten years. There are currently no clinical trials evaluating the efficacy or safety of Eltrombopag in second or further lines treatment of secondary ITP or Connettive Tissue Disease (CTD)-associated thrombocytopenia, with the unique exception of SLE. A few case reports showed that both Eltrombopag and Romiplostim can be successfully used in cases of SLE-associated thrombocytopenia, refractory to first-line immunosuppression therapy [58].

Some studies found an association between TPO-RA and incidence of VTE and arterial thrombosis in patients with chronic liver disease or acquired thrombophilia, such as aPL-positivity or APS. This may be due to the potential role of TPO-RA in enhancing the autoimmune response [59].

### 5.4. Hydroxychloroquine as a Possible Second Line Agent in APS-Associated Thrombocytopenia

HCQ is commonly adopted for the treatment of SLE. So far, only a few studies have explored the role of HCQ in the treatment of thrombocytopenia associated with SLE; in detail, combined treatment with prednisone and HCQ resulted effective in 7/11 (64%) patients among a cohort of 59 subjects affected by thrombocytopenia and SLE [60]. More recently, HCQ has been administered to treat thrombocytopenia after the failure of first-line oral prednisone treatment (N = 40), either in patients with SLE (N = 12) or with positive antinuclear antibodies (ANA), without overt SLE (N = 28) with a good response rate (60%), [61]. Interestingly, among the enrolled patients, LA, aPL, aCL, and antib2GP1 were also detected in the group without overt SLE. Based on these data, the administration of HCQ for the management of aPL-associated thrombocytopenia should be taken into account as a potential second-line option. Eular recommendations [62] have recently included in the “research agenda “, the evaluation of HCQ administration as a prophylactic treatment option in patients with high-risk aPL profile, including those with non criteria APS manifestations (i.e., thrombocytopenia). A study by Nuri et al. demonstrated that chronic therapy with HCQ can decrease aCL and anti-β2GPI IgG and IgM titers with seemingly no impact on thrombosis risk [63].

HCQ may also be able to prevent paradoxical clots by decreasing endothelial dysfunction induced by aPL [64].

## 6. Conclusions

Thrombocytopenia occurs in 20–50% of patients with a confirmed diagnosis of APS, but it is not considered as a defining criterion for the diagnosis of APS, and it is commonly not associated with significant bleeding risk. Thus, it is not currently part of the standard clinical assessment before prescribing long-term anticoagulation or LDA. Some studies, though, demonstrated that approximately 40% of patients with thrombocytopenia and APS had antibodies directed against the GPIIb-IIIa or GPIb-IX-V complexes, thus confirming the presence of antiplatelets antibodies in APS. Whether these antibodies can be assayed in the prognostic evaluation of patients with APS remains uncertain, although several studies have shown that thrombocytopenia can actually be regarded as a warning sign in the assessment of high-risk APS.

On the other hand, aPL were found in several patients with a confirmed diagnosis of ITP and associated with a thrombotic risk, not usually observed in ITP itself. Furthermore, it was demonstrated that the persistence of aPL in these subjects drastically reduced their overall survival due to an increased risk of paradoxical clots. It can be assumed that the evaluation of aPL and their persistent positivity during ITP should be periodically assessed to stratify the risk of thrombosis.

With reference to the prophylaxis and treatment of APS, the above-mentioned studies confirm the current guidelines that recommend long-term anticoagulation in patients with a high-risk APS and thrombotic events and the prophylaxis with LDA even in the presence of mild thrombocytopenia. It has been shown that immunosuppressive therapy, given to induce clinical remission in ITP, is also able to reduce aPL titers, with the notable mention of rituximab as one of the most effective drugs available in this setting. Specific long-term therapies for ITP (i.e., TPO-receptor agonists) should be carefully administered in persistently aPL-positive patients with chronic refractory ITP due to the risk of thrombotic events.

APS-associated thrombocytopenia and APS-associated ITP should be considered as similar conditions and treated in the same way. Up to date, thrombocytopenia cannot contribute to defining the diagnosis of APS, like the presence of aPL cannot be adopted to define the diagnosis of ITP, but it can be assumed that the evaluation of these parameters can support clinicians to identify a subgroup of high-risk patients where the best therapeutic approach should be carefully defined.

## Figures and Tables

**Figure 1 biomedicines-09-01170-f001:**
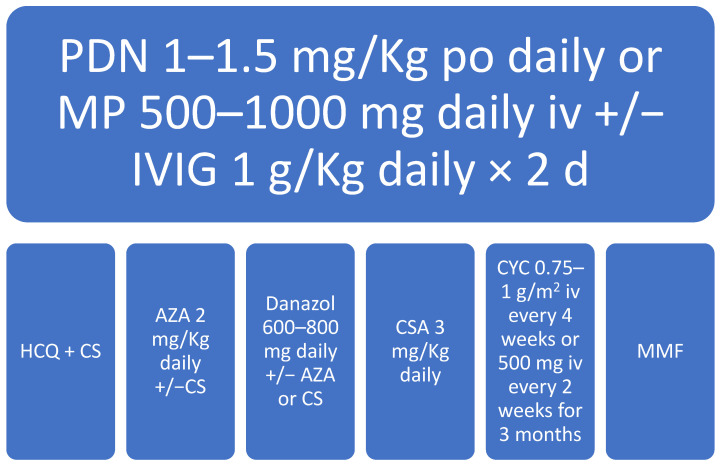
Algorithm for 1st and 2nd line treatment of SLE-associated thrombocytopenia. PDN: prednisone; MP: methylprednisolone; IVIG: intravenous immunoglobulins; HCQ: hydroxychloroquine; CS: corticosteroids; AZA: azathioprine; CSA: cyclosporine; CYC: cyclophosphamide; MMF: mycophenolate mofetil; d = days.

**Table 1 biomedicines-09-01170-t001:** Sapporo criteria for APS classification.

Clinical and Laboratory Criteria	
Vascular thrombosis	≥1 clinical episode of arterial, venous, or small vessel thrombosis. Thrombosis must be objectively confirmed. For histopathological confirmation, thrombosis must be present without inflammation of the vessel wall.
Pregnancy morbidity	≥1 unexplained death of a morphologically normal fetus ≥10 weeks of gestation. ≥1 premature delivery of a morphologically normal fetus <34 weeks of gestation because of. Severe preeclampsia or eclampsia defined according to standard definition. Recognized features of placental insufficiency. ≥3 unexplained consecutive miscarriages < 10 weeks of gestation, with maternal and paternal factors (anatomic, hormonal, or chromosomal abnormalities) excluded.
Laboratory criteria	Presence of aPL, on two or more occasions at least 12 weeks apart and no more than 5 years prior to clinical manifestations, as demonstrated by ≥1 of the following. LA. Medium to high-titer (>40 GPL or MPL, or >99th percentile) aCL IgG or IgM. Anti-β2 glycoprotein-I (anti-β2GPI) IgG or IgM > 99th percentile.

**Table 2 biomedicines-09-01170-t002:** Pathogenetic mechanisms involved in thrombocytopenia within the context of APS.

Pathogenesis Hypothesis	Pathway
Secondary Immune Thrombocytopenia	Expression of platelet membrane glycoproteins, especially GPIIb/IIIa, increases after aPL stimulation, and the binding of anti-β2GPI-β2GPI complex to receptors on the platelet membrane induces the activation and aggregation of platelets. Antibodies directed against GP on the cell wall of platelets (GPIIb/IIIa, GPIb/IX, GPIa/IIa, and GPIV) have been identified in 40%–70% of thrombocytopenic patients with APS.
Decreased platelet production	APS can associate with Hemophagocytic Syndrome, although it is extremely rare, and bone marrow necrosis. Both conditions may cause a decrease in platelet production.
Increased platelet pooling	This condition can be suspected in patients with splenomegaly secondary to portal vein or splenic vein thrombosis due to APS.
Increased platelet consumption	aPL can mediate upregulation of Von Willebrand Factor (vWF) production by endothelial cells and by direct platelet activation, resulting in an increased vWF-platelet binding.

**Table 3 biomedicines-09-01170-t003:** Potential treatments of APS with or without thrombosis.

Patient Group	Clinical History	I Line Therapy
non-triple-positive aPL carrier	No thrombotic events	No prophylaxis required
triple-positive aPL carrier	No thrombotic events	Primary prophylaxis with LDA may be considered
thrombotic APS	VTE	Secondary prophylaxis with LT-VKA (target INR 2.5, range 2–3)
thrombotic APS	Arterial thrombosis	Secondary prophylaxis with LT-VKA (target INR 3.5, range 3–4) or LDA
obstetric APS	VTE/arterial thrombosis	LDA + LMWH
CAPS	Without secondary CTD *	Anticoagulation + glucocorticoids + HD-IVIG + PEX
CAPS	Secondary CTD	Anticoagulation + glucocorticoids + HD-IVIG + PEX + cyclophosphamide
Thrombotic APS/CAPS	Refractory disease; Microangiopathic hemolytic anemia	Anticoagulation + glucocorticoids + HD-IVIG + PEX + rituximab

*: connettive tissue disease, including SLE.

## Data Availability

The data presented in this study are available on request from the corresponding author.

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
