# Peer review of "Immune Thrombocytopenia in Antiphospholipid Syndrome: Is It Primary or Secondary?"

_biomedicines, 2021, doi:10.3390/biomedicines9091170_

Round 1
Reviewer 1 Report
This review by Tomasello and colleagues addresses the pathogenic mechanisms and the clinical management of thrombocytopenia in anti-phospholipid syndrome. The topic is surely interesting, but the manuscript can be further improved.
- Line 42: The sentence “aPL represent the strongest factors involved in thrombotic risk within APS” is not clear. aPL are the serological markers of APS and exert a pathogenic role. Surely the thrombotic risk can be further stratified upon the aPL profile. Please rephrase.
- Line 48: platelet membrane glycoproteins UPDATED (?). Please edit.
- Line 50: Primary ITP is associated LEADS (?). Please edit.
- There is an inconsistent use of abbreviations in the text (ITP, aPL, b2GPI), please check.
- The link between thrombocytopenia, aPL and systemic lupus erythematosus should be discussed, from both a clinical and a therapeutic point of view.
- An additional mechanism explaining the link between thrombocytopenia and aPL might be the interaction of beta2GPI with von Willebrand factor. Anti-b2GPI antibodies neutralize this interaction, potentially accounting for decrease of platelet count.
- Thrombocytopenia is regarded as a “non criteria” manifestation of APS in the updated criteria for APS classification, please include this issue in the review.
- The evidence of thrombocytopenia in a subject with high risk persistent aPL positivity may prompt a treatment with LDA.
- Usually, the decision to institute a primary prophylaxis depends on the aPL profile: triple versus single aPL positivity. Please discuss.
- LDA has been proposed as treatment for aPL-positive patients with stroke, but there is general consensus that patients with arterial events deserve anticoagulation. Surely evidence is less strong for arterial than venous events, but some authors even propose a higher target INR for those with arterial events. Please rephrase this paragraph.
- Table 2: Patients with no event are usually called “asymptomatic aPL carriers”.
- Table 2: Please use the terms “thrombotic APS” for those with thrombotic events and “obstetric APS” for those with pregnancy morbidity.
- Table 2: Women with obstetric APS should receive LDA AND LMWH, please edit.
- Is there a role for hydroxychloroquine in the treatment of aPL-associated thrombocytopenia? Please discuss available evidence.
- Please define the cut-off for severe thrombocytopenia requiring “aggressive treatment”.
- Please provide definition for clinical response, and the timing for platelet count evaluation after treatment.
- Please provide indications for use of rituximab in clinical practice.
Author Response
Many thanks for your comments, we have now revised the manuscript according to your suggestions and we find it improved, please find below a point by point reply to your comments
This review by Tomasello and colleagues addresses the pathogenic mechanisms and the clinical management of thrombocytopenia in anti-phospholipid syndrome. The topic is surely interesting, but the manuscript can be further improved.
Line 42: The sentence “aPL represent the strongest factors involved in thrombotic risk within APS” is not clear. aPL are the serological markers of APS and exert a pathogenic role. Surely the thrombotic risk can be further stratified upon the aPL profile. Please rephrase.
Line 48: platelet membrane glycoproteins UPDATED (?). Please edit.
Line 50: Primary ITP is associated LEADS (?). Please edit.
There is an inconsistent use of abbreviations in the text (ITP, aPL, b2GPI), please check.
RE: Many thanks for your suggestions, we have now modified these syntax errors and deleted inappropriate terms
The link between thrombocytopenia, aPL and systemic lupus erythematosus should be discussed, from both a clinical and a therapeutic point of view.
RE: We have now added to our review a paragraph explaining this suggested link
An additional mechanism explaining the link between thrombocytopenia and aPL might be the interaction of beta2GPI with von Willebrand factor. Anti-b2GPI antibodies neutralize this interaction, potentially accounting for decrease of platelet count.
RE: This has now been added to Table 1 and reported in the reference list
Thrombocytopenia is regarded as a “non criteria” manifestation of APS in the updated criteria for APS classification, please include this issue in the review.
RE: This has now been added to the paragraph entitled “Diagnosis of APS”
The evidence of thrombocytopenia in a subject with high risk persistent aPL positivity may prompt a treatment with LDA. Usually, the decision to institute a primary prophylaxis depends on the aPL profile: triple versus single aPL positivity. Please discuss.
RE: This issue has now been treated in a dedicated paragraph within he section” An overview of the treatment of antiphospholipid antibodies syndrome : the latest guidelines”
LDA has been proposed as treatment for aPL-positive patients with stroke, but there is general consensus that patients with arterial events deserve anticoagulation. Surely evidence is less strong for arterial than venous events, but some authors even propose a higher target INR for those with arterial events. Please rephrase this paragraph.
RE: Many thanks the paragraph has now been rephrased
Table 2: Patients with no event are usually called “asymptomatic aPL carriers”.
Table 2: Please use the terms “thrombotic APS” for those with thrombotic events and “obstetric APS” for those with pregnancy morbidity.
Table 2: Women with obstetric APS should receive LDA AND LMWH, please edit.
RE: Tables have now been edited as suggested
Is there a role for hydroxychloroquine in the treatment of aPL-associated thrombocytopenia? Please discuss available evidence.
RE: This has now been treated in a dedicated paragraph
Please define the cut-off for severe thrombocytopenia requiring “aggressive treatment”.
RE: This cut-off value has now been added
Please provide definition for clinical response, and the timing for platelet count evaluation after treatment.
RE: This has now been added in a dedicated paragraph
Please provide indications for use of rituximab in clinical practice.
RE: This has been added to the paragraph on anti CD20 treatment
This review by Tomasello and colleagues addresses the pathogenic mechanisms and the clinical management of thrombocytopenia in anti-phospholipid syndrome. The topic is surely interesting, but the manuscript can be further improved.
Line 42: The sentence “aPL represent the strongest factors involved in thrombotic risk within APS” is not clear. aPL are the serological markers of APS and exert a pathogenic role. Surely the thrombotic risk can be further stratified upon the aPL profile. Please rephrase.
Line 48: platelet membrane glycoproteins UPDATED (?). Please edit.
Line 50: Primary ITP is associated LEADS (?). Please edit.
There is an inconsistent use of abbreviations in the text (ITP, aPL, b2GPI), please check.
RE: Many thanks for your suggestions, we have now modified these syntax errors and deleted inappropriate terms
The link between thrombocytopenia, aPL and systemic lupus erythematosus should be discussed, from both a clinical and a therapeutic point of view.
RE: We have now added to our review a paragraph explaining this suggested link
An additional mechanism explaining the link between thrombocytopenia and aPL might be the interaction of beta2GPI with von Willebrand factor. Anti-b2GPI antibodies neutralize this interaction, potentially accounting for decrease of platelet count.
RE: This has now been added to Table 1 and reported in the reference list
Thrombocytopenia is regarded as a “non criteria” manifestation of APS in the updated criteria for APS classification, please include this issue in the review.
RE: This has now been added to the paragraph entitled “Diagnosis of APS”
The evidence of thrombocytopenia in a subject with high risk persistent aPL positivity may prompt a treatment with LDA. Usually, the decision to institute a primary prophylaxis depends on the aPL profile: triple versus single aPL positivity. Please discuss.
RE: This issue has now been treated in a dedicated paragraph within he section” An overview of the treatment of antiphospholipid antibodies syndrome : the latest guidelines”
LDA has been proposed as treatment for aPL-positive patients with stroke, but there is general consensus that patients with arterial events deserve anticoagulation. Surely evidence is less strong for arterial than venous events, but some authors even propose a higher target INR for those with arterial events. Please rephrase this paragraph.
RE: Many thanks the paragraph has now been rephrased
Table 2: Patients with no event are usually called “asymptomatic aPL carriers”.
Table 2: Please use the terms “thrombotic APS” for those with thrombotic events and “obstetric APS” for those with pregnancy morbidity.
Table 2: Women with obstetric APS should receive LDA AND LMWH, please edit.
RE: Tables have now been edited as suggested
Is there a role for hydroxychloroquine in the treatment of aPL-associated thrombocytopenia? Please discuss available evidence.
RE: This has now been treated in a dedicated paragraph
Please define the cut-off for severe thrombocytopenia requiring “aggressive treatment”.
RE: This cut-off value has now been added
Please provide definition for clinical response, and the timing for platelet count evaluation after treatment.
RE: This has now been added in a dedicated paragraph
Please provide indications for use of rituximab in clinical practice.
RE: This has been added to the paragraph on anti CD20 treatment
This review by Tomasello and colleagues addresses the pathogenic mechanisms and the clinical management of thrombocytopenia in anti-phospholipid syndrome. The topic is surely interesting, but the manuscript can be further improved.
Line 42: The sentence “aPL represent the strongest factors involved in thrombotic risk within APS” is not clear. aPL are the serological markers of APS and exert a pathogenic role. Surely the thrombotic risk can be further stratified upon the aPL profile. Please rephrase.
Line 48: platelet membrane glycoproteins UPDATED (?). Please edit.
Line 50: Primary ITP is associated LEADS (?). Please edit.
There is an inconsistent use of abbreviations in the text (ITP, aPL, b2GPI), please check.
RE: Many thanks for your suggestions, we have now modified these syntax errors and deleted inappropriate terms
The link between thrombocytopenia, aPL and systemic lupus erythematosus should be discussed, from both a clinical and a therapeutic point of view.
RE: We have now added to our review a paragraph explaining this suggested link
An additional mechanism explaining the link between thrombocytopenia and aPL might be the interaction of beta2GPI with von Willebrand factor. Anti-b2GPI antibodies neutralize this interaction, potentially accounting for decrease of platelet count.
RE: This has now been added to Table 1 and reported in the reference list
Thrombocytopenia is regarded as a “non criteria” manifestation of APS in the updated criteria for APS classification, please include this issue in the review.
RE: This has now been added to the paragraph entitled “Diagnosis of APS”
The evidence of thrombocytopenia in a subject with high risk persistent aPL positivity may prompt a treatment with LDA. Usually, the decision to institute a primary prophylaxis depends on the aPL profile: triple versus single aPL positivity. Please discuss.
RE: This issue has now been treated in a dedicated paragraph within he section” An overview of the treatment of antiphospholipid antibodies syndrome : the latest guidelines”
LDA has been proposed as treatment for aPL-positive patients with stroke, but there is general consensus that patients with arterial events deserve anticoagulation. Surely evidence is less strong for arterial than venous events, but some authors even propose a higher target INR for those with arterial events. Please rephrase this paragraph.
RE: Many thanks the paragraph has now been rephrased
Table 2: Patients with no event are usually called “asymptomatic aPL carriers”.
Table 2: Please use the terms “thrombotic APS” for those with thrombotic events and “obstetric APS” for those with pregnancy morbidity.
Table 2: Women with obstetric APS should receive LDA AND LMWH, please edit.
RE: Tables have now been edited as suggested
Is there a role for hydroxychloroquine in the treatment of aPL-associated thrombocytopenia? Please discuss available evidence.
RE: This has now been treated in a dedicated paragraph
Please define the cut-off for severe thrombocytopenia requiring “aggressive treatment”.
RE: This cut-off value has now been added
Please provide definition for clinical response, and the timing for platelet count evaluation after treatment.
RE: This has now been added in a dedicated paragraph
Please provide indications for use of rituximab in clinical practice.
RE: This has been added to the paragraph on anti CD20 treatment
This review by Tomasello and colleagues addresses the pathogenic mechanisms and the clinical management of thrombocytopenia in anti-phospholipid syndrome. The topic is surely interesting, but the manuscript can be further improved.
Line 42: The sentence “aPL represent the strongest factors involved in thrombotic risk within APS” is not clear. aPL are the serological markers of APS and exert a pathogenic role. Surely the thrombotic risk can be further stratified upon the aPL profile. Please rephrase.
Line 48: platelet membrane glycoproteins UPDATED (?). Please edit.
Line 50: Primary ITP is associated LEADS (?). Please edit.
There is an inconsistent use of abbreviations in the text (ITP, aPL, b2GPI), please check.
RE: Many thanks for your suggestions, we have now modified these syntax errors and deleted inappropriate terms
The link between thrombocytopenia, aPL and systemic lupus erythematosus should be discussed, from both a clinical and a therapeutic point of view.
RE: We have now added to our review a paragraph explaining this suggested link
An additional mechanism explaining the link between thrombocytopenia and aPL might be the interaction of beta2GPI with von Willebrand factor. Anti-b2GPI antibodies neutralize this interaction, potentially accounting for decrease of platelet count.
RE: This has now been added to Table 1 and reported in the reference list
Thrombocytopenia is regarded as a “non criteria” manifestation of APS in the updated criteria for APS classification, please include this issue in the review.
RE: This has now been added to the paragraph entitled “Diagnosis of APS”
The evidence of thrombocytopenia in a subject with high risk persistent aPL positivity may prompt a treatment with LDA. Usually, the decision to institute a primary prophylaxis depends on the aPL profile: triple versus single aPL positivity. Please discuss.
RE: This issue has now been treated in a dedicated paragraph within he section” An overview of the treatment of antiphospholipid antibodies syndrome : the latest guidelines”
LDA has been proposed as treatment for aPL-positive patients with stroke, but there is general consensus that patients with arterial events deserve anticoagulation. Surely evidence is less strong for arterial than venous events, but some authors even propose a higher target INR for those with arterial events. Please rephrase this paragraph.
RE: Many thanks the paragraph has now been rephrased
Table 2: Patients with no event are usually called “asymptomatic aPL carriers”.
Table 2: Please use the terms “thrombotic APS” for those with thrombotic events and “obstetric APS” for those with pregnancy morbidity.
Table 2: Women with obstetric APS should receive LDA AND LMWH, please edit.
RE: Tables have now been edited as suggested
Is there a role for hydroxychloroquine in the treatment of aPL-associated thrombocytopenia? Please discuss available evidence.
RE: This has now been treated in a dedicated paragraph
Please define the cut-off for severe thrombocytopenia requiring “aggressive treatment”.
RE: This cut-off value has now been added
Please provide definition for clinical response, and the timing for platelet count evaluation after treatment.
RE: This has now been added in a dedicated paragraph
Please provide indications for use of rituximab in clinical practice.
RE: This has been added to the paragraph on anti CD20 treatment
Reviewer 2 Report
Very nicely done review. I only have a comment on the chapter
2. APS and Thrombocytopenia,
where it is listed in Table 1 among the pathogenesis hypothesis of thrombocytopenia - pseudothrombocytopenia. Pseudothrombocytopenia is only laboratory phenomenon not true thrombocytopenia. I suggest taking this into account in the table.
Author Response
We thank you very much for your positive evaluation of the paper, please find below a point by point reply to your comments:
Very nicely done review. I only have a comment on the chapter
RE: Many thanks for your high consideration of our work
- APS and Thrombocytopenia,
where it is listed in Table 1 among the pathogenesis hypothesis of thrombocytopenia - pseudothrombocytopenia. Pseudothrombocytopenia is only laboratory phenomenon not true thrombocytopenia. I suggest taking this into account in the table.
RE: We have now deleted “pseudothrombocytopenia” from the table .
Very nicely done review. I only have a comment on the chapter
RE: Many thanks for your high consideration of our work
- APS and Thrombocytopenia,
where it is listed in Table 1 among the pathogenesis hypothesis of thrombocytopenia - pseudothrombocytopenia. Pseudothrombocytopenia is only laboratory phenomenon not true thrombocytopenia. I suggest taking this into account in the table.
RE: We have now deleted “pseudothrombocytopenia” from the table .
Very nicely done review. I only have a comment on the chapter
RE: Many thanks for your high consideration of our work
- APS and Thrombocytopenia,
where it is listed in Table 1 among the pathogenesis hypothesis of thrombocytopenia - pseudothrombocytopenia. Pseudothrombocytopenia is only laboratory phenomenon not true thrombocytopenia. I suggest taking this into account in the table.
RE: We have now deleted “pseudothrombocytopenia” from the table .
Very nicely done review. I only have a comment on the chapter
RE: Many thanks for your high consideration of our work
- APS and Thrombocytopenia,
where it is listed in Table 1 among the pathogenesis hypothesis of thrombocytopenia - pseudothrombocytopenia. Pseudothrombocytopenia is only laboratory phenomenon not true thrombocytopenia. I suggest taking this into account in the table.
RE: We have now deleted “pseudothrombocytopenia” from the table .
Very nicely done review. I only have a comment on the chapter
RE: Many thanks for your high consideration of our work
- APS and Thrombocytopenia,
where it is listed in Table 1 among the pathogenesis hypothesis of thrombocytopenia - pseudothrombocytopenia. Pseudothrombocytopenia is only laboratory phenomenon not true thrombocytopenia. I suggest taking this into account in the table.
RE: We have now deleted “pseudothrombocytopenia” from the table .
Very nicely done review. I only have a comment on the chapter
RE: Many thanks for your high consideration of our work
- APS and Thrombocytopenia,
where it is listed in Table 1 among the pathogenesis hypothesis of thrombocytopenia - pseudothrombocytopenia. Pseudothrombocytopenia is only laboratory phenomenon not true thrombocytopenia. I suggest taking this into account in the table.
RE: We have now deleted “pseudothrombocytopenia” from the table .
Very nicely done review. I only have a comment on the chapter
RE: Many thanks for your high consideration of our work
- APS and Thrombocytopenia,
where it is listed in Table 1 among the pathogenesis hypothesis of thrombocytopenia - pseudothrombocytopenia. Pseudothrombocytopenia is only laboratory phenomenon not true thrombocytopenia. I suggest taking this into account in the table.
RE: We have now deleted “pseudothrombocytopenia” from the table .
Very nicely done review. I only have a comment on the chapter
RE: Many thanks for your high consideration of our work
- APS and Thrombocytopenia,
where it is listed in Table 1 among the pathogenesis hypothesis of thrombocytopenia - pseudothrombocytopenia. Pseudothrombocytopenia is only laboratory phenomenon not true thrombocytopenia. I suggest taking this into account in the table.
RE: We have now deleted “pseudothrombocytopenia” from the table .
Very nicely done review. I only have a comment on the chapter
RE: Many thanks for your high consideration of our work
- APS and Thrombocytopenia,
where it is listed in Table 1 among the pathogenesis hypothesis of thrombocytopenia - pseudothrombocytopenia. Pseudothrombocytopenia is only laboratory phenomenon not true thrombocytopenia. I suggest taking this into account in the table.
RE: We have now deleted “pseudothrombocytopenia” from the table .
Round 2
Reviewer 1 Report
The authors have assessed the criticisms raised by the referee.
However, there are still some issue that should be addressed.
The main point is the inconsistent use of acronyms, it seems that the paper has not been checked. The reader gets lost!
- The acronym for anti-phospholipid antibodies has been inserted in line 36, and is aPL. Throughout the manuscript, one can read aPLs, aPL antibodies (line 130, 135, 159), again anti-phospholipid antibodies in extenso (e.g. line 86, 87)
- The acronym for anti-cardiolipin antibodies has been inserted in line 36, and is aCL. Throughout the manuscript, one can read aCL, aCA,magain anti-cardiolipin antibodies in extenso.
- The same for LA.
- Please refer to low dose acetyl salycilic acid and non aspirin. Again, please be consistent with acronyms (LDA, LD-ASA).
- please insert the abbreviation the first time you cite it, e.g. for HCQ.
- HydroXychloroquine (line 319)
- Sometimes the text is repetitive (and even conflicting):
- e.g. line 185: please erase the sentence “According to ISTH guidelines, an aPL profile can be regarded as responsible…”
- LDA primary prophylaxis: although there is no evidence from RCT, in patients with high risk profile (double/triple aPL positivity) it is common practice to prescribe primary thromboprophylaxis with LDA. Please be consistent (line 228 versus line 245)
- Line 273: Pregnant women with thrombotic APS should be treated. Please erase “or a history of VTE and/or arterial thrombosis related to aPL: this is thrombotic APS)
- Please do not include management of carriers in Table 2, as this is debatable.
Line 59: Please add non criteria MANIFESTATION
Line 59: Please rephrase to “This is a still gray area, indeed in the Task Force Report….”
Line 330: Please cite RITAPS study for rituximab in APS.
Author Response
We thank reviewer for the precious support, we have made the required minor revisions and we are here reporting a point by point reply to the comments
The authors have assessed the criticisms raised by the referee.However, there are still some issue that should be addressed.The main point is the inconsistent use of acronyms, it seems that the paper has not been checked. The reader gets lost!
We have now cheched more accurately for the inconsistent use of abbreviations the manuscript
- The acronym for anti-phospholipid antibodies has been inserted in line 36, and is aPL. Throughout the manuscript, one can read aPLs, aPL antibodies (line 130, 135, 159), again anti-phospholipid antibodies in extenso (e.g. line 86, 87)
RE: Many thanks, we have now modified the manuscript reporting only the acronym aPL after the first citation of it in line 36
- The acronym for anti-cardiolipin antibodies has been inserted in line 36, and is aCL. Throughout the manuscript, one can read aCL, aCA,magain anti-cardiolipin antibodies in extenso.
RE: Many thanks, we have now modified the manuscript reporting only the acronym aCL after the first citation of it in line 36
- The same for LA.
RE: We have now adopted only LA after the first citation of Lupus Anticoagulant in the text
- Please refer to low dose acetyl salycilic acid and non aspirin. Again, please be consistent with acronyms (LDA, LD-ASA).
RE: We have now adopted the term low dose acetyl salycilic acid and its acronym (LDA) and replaced it within the text
- please insert the abbreviation the first time you cite it, e.g. for HCQ.
This has now been inserted, and HCQ has been replaced the first time the term Hydroxychloroquine has been adopted
- HydroXychloroquine (line 319)
This term has now been modified,thank you
- Sometimes the text is repetitive (and even conflicting):
- e.g. line 185: please erase the sentence “According to ISTH guidelines, an aPL profile can be regarded as responsible…”
Thank you this sentence has now been deleted
- LDA primary prophylaxis: although there is no evidence from RCT, in patients with high risk profile (double/triple aPL positivity) it is common practice to prescribe primary thromboprophylaxis with LDA. Please be consistent (line 228 versus line 245)
RE: We have now modified the first sentence reporting” however this practice is not supported by evidences from randomized controlled trials”, this makes the following sentence consistent
“The protective role of LDA for the primary prophylaxis of thrombosis is, uptodate, not supported by robust evidences, also for those patients with high risk persistent aPL positivity. Primary prevention of thrombosis should be thus defined on an individual basis in patients with high risk aPL, It should be based on regular assay of antibodies titer and control of any other modifiable risk factor for thrombosis, along with treatment of any underlying known autoimmune disease [43], whether or not within this context, risk strati-fication may include also additional “non criteria” manifestations like thrombocytopenia needs to be clarified yet
- Line 273: Pregnant women with thrombotic APS should be treated. Please erase “or a history of VTE and/or arterial thrombosis related to aPL: this is thrombotic APS)
RE: Thank you, this sentence has now been deleted
- Please do not include management of carriers in Table 2, as this is debatable.
RE: Thank you, this line has now been deleted in Table 2
Line 59: Please add non criteria MANIFESTATION
RE: Thank you, the term manifestation has now been added
Line 59: Please rephrase to “This is a still gray area, indeed in the Task Force Report….”
RE: Thank you, this sentence has now been rephrased this way”: this is however still debated, ,the last reported meeting of the Technical Task Force Report on Antiphospholipid Syndrome Clinical Features, suggested in fact to consider thrombocytopenia as a classification criterion of APS [13].
Line 330: Please cite RITAPS study for rituximab in APS.
RE: Many thanks, this has now been added to text and references
The authors have assessed the criticisms raised by the referee.
However, there are still some issue that should be addressed.
The main point is the inconsistent use of acronyms, it seems that the paper has not been checked. The reader gets lost!
- The acronym for anti-phospholipid antibodies has been inserted in line 36, and is aPL. Throughout the manuscript, one can read aPLs, aPL antibodies (line 130, 135, 159), again anti-phospholipid antibodies in extenso (e.g. line 86, 87)
RE: Many thanks, we have now modified the manuscript reporting only the acronym aPL after the first citation of it in line 36
- The acronym for anti-cardiolipin antibodies has been inserted in line 36, and is aCL. Throughout the manuscript, one can read aCL, aCA,magain anti-cardiolipin antibodies in extenso.
RE: Many thanks, we have now modified the manuscript reporting only the acronym aCL after the first citation of it in line 36
- The same for LA.
RE: We have now adopted only LA after the first citation of Lupus Anticoagulant in the text
- Please refer to low dose acetyl salycilic acid and non aspirin. Again, please be consistent with acronyms (LDA, LD-ASA).
RE: We have now adopted the term low dose acetyl salycilic acid and its acronym (LDA) and replaced it within the text
- please insert the abbreviation the first time you cite it, e.g. for HCQ.
This has now been inserted, and HCQ has been replaced the first time the term Hydroxychloroquine has been adopted
- HydroXychloroquine (line 319)
This term has now been modified,thank you
- Sometimes the text is repetitive (and even conflicting):
- e.g. line 185: please erase the sentence “According to ISTH guidelines, an aPL profile can be regarded as responsible…”
Thank you this sentence has now been deleted
- LDA primary prophylaxis: although there is no evidence from RCT, in patients with high risk profile (double/triple aPL positivity) it is common practice to prescribe primary thromboprophylaxis with LDA. Please be consistent (line 228 versus line 245)
RE: We have now modified the first sentence reporting” however this practice is not supported by evidences from randomized controlled trials”, this makes the following sentence consistent
“The protective role of LDA for the primary prophylaxis of thrombosis is, uptodate, not supported by robust evidences, also for those patients with high risk persistent aPL positivity. Primary prevention of thrombosis should be thus defined on an individual basis in patients with high risk aPL, It should be based on regular assay of antibodies titer and control of any other modifiable risk factor for thrombosis, along with treatment of any underlying known autoimmune disease [43], whether or not within this context, risk strati-fication may include also additional “non criteria” manifestations like thrombocytopenia needs to be clarified yet
- Line 273: Pregnant women with thrombotic APS should be treated. Please erase “or a history of VTE and/or arterial thrombosis related to aPL: this is thrombotic APS)
RE: Thank you, this sentence has now been deleted
- Please do not include management of carriers in Table 2, as this is debatable.
RE: Thank you, this line has now been deleted in Table 2
Line 59: Please add non criteria MANIFESTATION
RE: Thank you, the term manifestation has now been added
Line 59: Please rephrase to “This is a still gray area, indeed in the Task Force Report….”
RE: Thank you, this sentence has now been rephrased this way”: this is however still debated, ,the last reported meeting of the Technical Task Force Report on Antiphospholipid Syndrome Clinical Features, suggested in fact to consider thrombocytopenia as a classification criterion of APS [13].
Line 330: Please cite RITAPS study for rituximab in APS.
RE: Many thanks, this has now been added to text and references
The authors have assessed the criticisms raised by the referee.
However, there are still some issue that should be addressed.
The main point is the inconsistent use of acronyms, it seems that the paper has not been checked. The reader gets lost!
- The acronym for anti-phospholipid antibodies has been inserted in line 36, and is aPL. Throughout the manuscript, one can read aPLs, aPL antibodies (line 130, 135, 159), again anti-phospholipid antibodies in extenso (e.g. line 86, 87)
RE: Many thanks, we have now modified the manuscript reporting only the acronym aPL after the first citation of it in line 36
- The acronym for anti-cardiolipin antibodies has been inserted in line 36, and is aCL. Throughout the manuscript, one can read aCL, aCA,magain anti-cardiolipin antibodies in extenso.
RE: Many thanks, we have now modified the manuscript reporting only the acronym aCL after the first citation of it in line 36
- The same for LA.
RE: We have now adopted only LA after the first citation of Lupus Anticoagulant in the text
- Please refer to low dose acetyl salycilic acid and non aspirin. Again, please be consistent with acronyms (LDA, LD-ASA).
RE: We have now adopted the term low dose acetyl salycilic acid and its acronym (LDA) and replaced it within the text
- please insert the abbreviation the first time you cite it, e.g. for HCQ.
This has now been inserted, and HCQ has been replaced the first time the term Hydroxychloroquine has been adopted
- HydroXychloroquine (line 319)
This term has now been modified,thank you
- Sometimes the text is repetitive (and even conflicting):
- e.g. line 185: please erase the sentence “According to ISTH guidelines, an aPL profile can be regarded as responsible…”
Thank you this sentence has now been deleted
- LDA primary prophylaxis: although there is no evidence from RCT, in patients with high risk profile (double/triple aPL positivity) it is common practice to prescribe primary thromboprophylaxis with LDA. Please be consistent (line 228 versus line 245)
RE: We have now modified the first sentence reporting” however this practice is not supported by evidences from randomized controlled trials”, this makes the following sentence consistent
“The protective role of LDA for the primary prophylaxis of thrombosis is, uptodate, not supported by robust evidences, also for those patients with high risk persistent aPL positivity. Primary prevention of thrombosis should be thus defined on an individual basis in patients with high risk aPL, It should be based on regular assay of antibodies titer and control of any other modifiable risk factor for thrombosis, along with treatment of any underlying known autoimmune disease [43], whether or not within this context, risk strati-fication may include also additional “non criteria” manifestations like thrombocytopenia needs to be clarified yet
- Line 273: Pregnant women with thrombotic APS should be treated. Please erase “or a history of VTE and/or arterial thrombosis related to aPL: this is thrombotic APS)
RE: Thank you, this sentence has now been deleted
- Please do not include management of carriers in Table 2, as this is debatable.
RE: Thank you, this line has now been deleted in Table 2
Line 59: Please add non criteria MANIFESTATION
RE: Thank you, the term manifestation has now been added
Line 59: Please rephrase to “This is a still gray area, indeed in the Task Force Report….”
RE: Thank you, this sentence has now been rephrased this way”: this is however still debated, ,the last reported meeting of the Technical Task Force Report on Antiphospholipid Syndrome Clinical Features, suggested in fact to consider thrombocytopenia as a classification criterion of APS [13].
Line 330: Please cite RITAPS study for rituximab in APS.
RE: Many thanks, this has now been added to text and references
The authors have assessed the criticisms raised by the referee.
However, there are still some issue that should be addressed.
The main point is the inconsistent use of acronyms, it seems that the paper has not been checked. The reader gets lost!
- The acronym for anti-phospholipid antibodies has been inserted in line 36, and is aPL. Throughout the manuscript, one can read aPLs, aPL antibodies (line 130, 135, 159), again anti-phospholipid antibodies in extenso (e.g. line 86, 87)
RE: Many thanks, we have now modified the manuscript reporting only the acronym aPL after the first citation of it in line 36
- The acronym for anti-cardiolipin antibodies has been inserted in line 36, and is aCL. Throughout the manuscript, one can read aCL, aCA,magain anti-cardiolipin antibodies in extenso.
RE: Many thanks, we have now modified the manuscript reporting only the acronym aCL after the first citation of it in line 36
- The same for LA.
RE: We have now adopted only LA after the first citation of Lupus Anticoagulant in the text
- Please refer to low dose acetyl salycilic acid and non aspirin. Again, please be consistent with acronyms (LDA, LD-ASA).
RE: We have now adopted the term low dose acetyl salycilic acid and its acronym (LDA) and replaced it within the text
- please insert the abbreviation the first time you cite it, e.g. for HCQ.
This has now been inserted, and HCQ has been replaced the first time the term Hydroxychloroquine has been adopted
- HydroXychloroquine (line 319)
This term has now been modified,thank you
- Sometimes the text is repetitive (and even conflicting):
- e.g. line 185: please erase the sentence “According to ISTH guidelines, an aPL profile can be regarded as responsible…”
Thank you this sentence has now been deleted
- LDA primary prophylaxis: although there is no evidence from RCT, in patients with high risk profile (double/triple aPL positivity) it is common practice to prescribe primary thromboprophylaxis with LDA. Please be consistent (line 228 versus line 245)
RE: We have now modified the first sentence reporting” however this practice is not supported by evidences from randomized controlled trials”, this makes the following sentence consistent
“The protective role of LDA for the primary prophylaxis of thrombosis is, uptodate, not supported by robust evidences, also for those patients with high risk persistent aPL positivity. Primary prevention of thrombosis should be thus defined on an individual basis in patients with high risk aPL, It should be based on regular assay of antibodies titer and control of any other modifiable risk factor for thrombosis, along with treatment of any underlying known autoimmune disease [43], whether or not within this context, risk strati-fication may include also additional “non criteria” manifestations like thrombocytopenia needs to be clarified yet
- Line 273: Pregnant women with thrombotic APS should be treated. Please erase “or a history of VTE and/or arterial thrombosis related to aPL: this is thrombotic APS)
RE: Thank you, this sentence has now been deleted
- Please do not include management of carriers in Table 2, as this is debatable.
RE: Thank you, this line has now been deleted in Table 2
Line 59: Please add non criteria MANIFESTATION
RE: Thank you, the term manifestation has now been added
Line 59: Please rephrase to “This is a still gray area, indeed in the Task Force Report….”
RE: Thank you, this sentence has now been rephrased this way”: this is however still debated, ,the last reported meeting of the Technical Task Force Report on Antiphospholipid Syndrome Clinical Features, suggested in fact to consider thrombocytopenia as a classification criterion of APS [13].
Line 330: Please cite RITAPS study for rituximab in APS.
RE: Many thanks, this has now been added to text and references
The authors have assessed the criticisms raised by the referee.
However, there are still some issue that should be addressed.
The main point is the inconsistent use of acronyms, it seems that the paper has not been checked. The reader gets lost!
- The acronym for anti-phospholipid antibodies has been inserted in line 36, and is aPL. Throughout the manuscript, one can read aPLs, aPL antibodies (line 130, 135, 159), again anti-phospholipid antibodies in extenso (e.g. line 86, 87)
RE: Many thanks, we have now modified the manuscript reporting only the acronym aPL after the first citation of it in line 36
- The acronym for anti-cardiolipin antibodies has been inserted in line 36, and is aCL. Throughout the manuscript, one can read aCL, aCA,magain anti-cardiolipin antibodies in extenso.
RE: Many thanks, we have now modified the manuscript reporting only the acronym aCL after the first citation of it in line 36
- The same for LA.
RE: We have now adopted only LA after the first citation of Lupus Anticoagulant in the text
- Please refer to low dose acetyl salycilic acid and non aspirin. Again, please be consistent with acronyms (LDA, LD-ASA).
RE: We have now adopted the term low dose acetyl salycilic acid and its acronym (LDA) and replaced it within the text
- please insert the abbreviation the first time you cite it, e.g. for HCQ.
This has now been inserted, and HCQ has been replaced the first time the term Hydroxychloroquine has been adopted
- HydroXychloroquine (line 319)
This term has now been modified,thank you
- Sometimes the text is repetitive (and even conflicting):
- e.g. line 185: please erase the sentence “According to ISTH guidelines, an aPL profile can be regarded as responsible…”
Thank you this sentence has now been deleted
- LDA primary prophylaxis: although there is no evidence from RCT, in patients with high risk profile (double/triple aPL positivity) it is common practice to prescribe primary thromboprophylaxis with LDA. Please be consistent (line 228 versus line 245)
RE: We have now modified the first sentence reporting” however this practice is not supported by evidences from randomized controlled trials”, this makes the following sentence consistent
“The protective role of LDA for the primary prophylaxis of thrombosis is, uptodate, not supported by robust evidences, also for those patients with high risk persistent aPL positivity. Primary prevention of thrombosis should be thus defined on an individual basis in patients with high risk aPL, It should be based on regular assay of antibodies titer and control of any other modifiable risk factor for thrombosis, along with treatment of any underlying known autoimmune disease [43], whether or not within this context, risk strati-fication may include also additional “non criteria” manifestations like thrombocytopenia needs to be clarified yet
- Line 273: Pregnant women with thrombotic APS should be treated. Please erase “or a history of VTE and/or arterial thrombosis related to aPL: this is thrombotic APS)
RE: Thank you, this sentence has now been deleted
- Please do not include management of carriers in Table 2, as this is debatable.
RE: Thank you, this line has now been deleted in Table 2
Line 59: Please add non criteria MANIFESTATION
RE: Thank you, the term manifestation has now been added
Line 59: Please rephrase to “This is a still gray area, indeed in the Task Force Report….”
RE: Thank you, this sentence has now been rephrased this way”: this is however still debated, ,the last reported meeting of the Technical Task Force Report on Antiphospholipid Syndrome Clinical Features, suggested in fact to consider thrombocytopenia as a classification criterion of APS [13].
Line 330: Please cite RITAPS study for rituximab in APS.
RE: Many thanks, this has now been added to text and references
The authors have assessed the criticisms raised by the referee.
However, there are still some issue that should be addressed.
The main point is the inconsistent use of acronyms, it seems that the paper has not been checked. The reader gets lost!
- The acronym for anti-phospholipid antibodies has been inserted in line 36, and is aPL. Throughout the manuscript, one can read aPLs, aPL antibodies (line 130, 135, 159), again anti-phospholipid antibodies in extenso (e.g. line 86, 87)
RE: Many thanks, we have now modified the manuscript reporting only the acronym aPL after the first citation of it in line 36
- The acronym for anti-cardiolipin antibodies has been inserted in line 36, and is aCL. Throughout the manuscript, one can read aCL, aCA,magain anti-cardiolipin antibodies in extenso.
RE: Many thanks, we have now modified the manuscript reporting only the acronym aCL after the first citation of it in line 36
- The same for LA.
RE: We have now adopted only LA after the first citation of Lupus Anticoagulant in the text
- Please refer to low dose acetyl salycilic acid and non aspirin. Again, please be consistent with acronyms (LDA, LD-ASA).
RE: We have now adopted the term low dose acetyl salycilic acid and its acronym (LDA) and replaced it within the text
- please insert the abbreviation the first time you cite it, e.g. for HCQ.
This has now been inserted, and HCQ has been replaced the first time the term Hydroxychloroquine has been adopted
- HydroXychloroquine (line 319)
This term has now been modified,thank you
- Sometimes the text is repetitive (and even conflicting):
- e.g. line 185: please erase the sentence “According to ISTH guidelines, an aPL profile can be regarded as responsible…”
Thank you this sentence has now been deleted
- LDA primary prophylaxis: although there is no evidence from RCT, in patients with high risk profile (double/triple aPL positivity) it is common practice to prescribe primary thromboprophylaxis with LDA. Please be consistent (line 228 versus line 245)
RE: We have now modified the first sentence reporting” however this practice is not supported by evidences from randomized controlled trials”, this makes the following sentence consistent
“The protective role of LDA for the primary prophylaxis of thrombosis is, uptodate, not supported by robust evidences, also for those patients with high risk persistent aPL positivity. Primary prevention of thrombosis should be thus defined on an individual basis in patients with high risk aPL, It should be based on regular assay of antibodies titer and control of any other modifiable risk factor for thrombosis, along with treatment of any underlying known autoimmune disease [43], whether or not within this context, risk strati-fication may include also additional “non criteria” manifestations like thrombocytopenia needs to be clarified yet
- Line 273: Pregnant women with thrombotic APS should be treated. Please erase “or a history of VTE and/or arterial thrombosis related to aPL: this is thrombotic APS)
RE: Thank you, this sentence has now been deleted
- Please do not include management of carriers in Table 2, as this is debatable.
RE: Thank you, this line has now been deleted in Table 2
Line 59: Please add non criteria MANIFESTATION
RE: Thank you, the term manifestation has now been added
Line 59: Please rephrase to “This is a still gray area, indeed in the Task Force Report….”
RE: Thank you, this sentence has now been rephrased this way”: this is however still debated, ,the last reported meeting of the Technical Task Force Report on Antiphospholipid Syndrome Clinical Features, suggested in fact to consider thrombocytopenia as a classification criterion of APS [13].
Line 330: Please cite RITAPS study for rituximab in APS.
RE: Many thanks, this has now been added to text and references
The authors have assessed the criticisms raised by the referee.
However, there are still some issue that should be addressed.
The main point is the inconsistent use of acronyms, it seems that the paper has not been checked. The reader gets lost!
- The acronym for anti-phospholipid antibodies has been inserted in line 36, and is aPL. Throughout the manuscript, one can read aPLs, aPL antibodies (line 130, 135, 159), again anti-phospholipid antibodies in extenso (e.g. line 86, 87)
RE: Many thanks, we have now modified the manuscript reporting only the acronym aPL after the first citation of it in line 36
- The acronym for anti-cardiolipin antibodies has been inserted in line 36, and is aCL. Throughout the manuscript, one can read aCL, aCA,magain anti-cardiolipin antibodies in extenso.
RE: Many thanks, we have now modified the manuscript reporting only the acronym aCL after the first citation of it in line 36
- The same for LA.
RE: We have now adopted only LA after the first citation of Lupus Anticoagulant in the text
- Please refer to low dose acetyl salycilic acid and non aspirin. Again, please be consistent with acronyms (LDA, LD-ASA).
RE: We have now adopted the term low dose acetyl salycilic acid and its acronym (LDA) and replaced it within the text
- please insert the abbreviation the first time you cite it, e.g. for HCQ.
This has now been inserted, and HCQ has been replaced the first time the term Hydroxychloroquine has been adopted
- HydroXychloroquine (line 319)
This term has now been modified,thank you
- Sometimes the text is repetitive (and even conflicting):
- e.g. line 185: please erase the sentence “According to ISTH guidelines, an aPL profile can be regarded as responsible…”
Thank you this sentence has now been deleted
- LDA primary prophylaxis: although there is no evidence from RCT, in patients with high risk profile (double/triple aPL positivity) it is common practice to prescribe primary thromboprophylaxis with LDA. Please be consistent (line 228 versus line 245)
RE: We have now modified the first sentence reporting” however this practice is not supported by evidences from randomized controlled trials”, this makes the following sentence consistent
“The protective role of LDA for the primary prophylaxis of thrombosis is, uptodate, not supported by robust evidences, also for those patients with high risk persistent aPL positivity. Primary prevention of thrombosis should be thus defined on an individual basis in patients with high risk aPL, It should be based on regular assay of antibodies titer and control of any other modifiable risk factor for thrombosis, along with treatment of any underlying known autoimmune disease [43], whether or not within this context, risk strati-fication may include also additional “non criteria” manifestations like thrombocytopenia needs to be clarified yet
- Line 273: Pregnant women with thrombotic APS should be treated. Please erase “or a history of VTE and/or arterial thrombosis related to aPL: this is thrombotic APS)
RE: Thank you, this sentence has now been deleted
- Please do not include management of carriers in Table 2, as this is debatable.
RE: Thank you, this line has now been deleted in Table 2
Line 59: Please add non criteria MANIFESTATION
RE: Thank you, the term manifestation has now been added
Line 59: Please rephrase to “This is a still gray area, indeed in the Task Force Report….”
RE: Thank you, this sentence has now been rephrased this way”: this is however still debated, ,the last reported meeting of the Technical Task Force Report on Antiphospholipid Syndrome Clinical Features, suggested in fact to consider thrombocytopenia as a classification criterion of APS [13].
Line 330: Please cite RITAPS study for rituximab in APS.
RE: Many thanks, this has now been added to text and references
The authors have assessed the criticisms raised by the referee.
However, there are still some issue that should be addressed.
The main point is the inconsistent use of acronyms, it seems that the paper has not been checked. The reader gets lost!
- The acronym for anti-phospholipid antibodies has been inserted in line 36, and is aPL. Throughout the manuscript, one can read aPLs, aPL antibodies (line 130, 135, 159), again anti-phospholipid antibodies in extenso (e.g. line 86, 87)
RE: Many thanks, we have now modified the manuscript reporting only the acronym aPL after the first citation of it in line 36
- The acronym for anti-cardiolipin antibodies has been inserted in line 36, and is aCL. Throughout the manuscript, one can read aCL, aCA,magain anti-cardiolipin antibodies in extenso.
RE: Many thanks, we have now modified the manuscript reporting only the acronym aCL after the first citation of it in line 36
- The same for LA.
RE: We have now adopted only LA after the first citation of Lupus Anticoagulant in the text
- Please refer to low dose acetyl salycilic acid and non aspirin. Again, please be consistent with acronyms (LDA, LD-ASA).
RE: We have now adopted the term low dose acetyl salycilic acid and its acronym (LDA) and replaced it within the text
- please insert the abbreviation the first time you cite it, e.g. for HCQ.
This has now been inserted, and HCQ has been replaced the first time the term Hydroxychloroquine has been adopted
- HydroXychloroquine (line 319)
This term has now been modified,thank you
- Sometimes the text is repetitive (and even conflicting):
- e.g. line 185: please erase the sentence “According to ISTH guidelines, an aPL profile can be regarded as responsible…”
Thank you this sentence has now been deleted
- LDA primary prophylaxis: although there is no evidence from RCT, in patients with high risk profile (double/triple aPL positivity) it is common practice to prescribe primary thromboprophylaxis with LDA. Please be consistent (line 228 versus line 245)
RE: We have now modified the first sentence reporting” however this practice is not supported by evidences from randomized controlled trials”, this makes the following sentence consistent
“The protective role of LDA for the primary prophylaxis of thrombosis is, uptodate, not supported by robust evidences, also for those patients with high risk persistent aPL positivity. Primary prevention of thrombosis should be thus defined on an individual basis in patients with high risk aPL, It should be based on regular assay of antibodies titer and control of any other modifiable risk factor for thrombosis, along with treatment of any underlying known autoimmune disease [43], whether or not within this context, risk strati-fication may include also additional “non criteria” manifestations like thrombocytopenia needs to be clarified yet
- Line 273: Pregnant women with thrombotic APS should be treated. Please erase “or a history of VTE and/or arterial thrombosis related to aPL: this is thrombotic APS)
RE: Thank you, this sentence has now been deleted
- Please do not include management of carriers in Table 2, as this is debatable.
RE: Thank you, this line has now been deleted in Table 2
Line 59: Please add non criteria MANIFESTATION
RE: Thank you, the term manifestation has now been added
Line 59: Please rephrase to “This is a still gray area, indeed in the Task Force Report….”
RE: Thank you, this sentence has now been rephrased this way”: this is however still debated, ,the last reported meeting of the Technical Task Force Report on Antiphospholipid Syndrome Clinical Features, suggested in fact to consider thrombocytopenia as a classification criterion of APS [13].
Line 330: Please cite RITAPS study for rituximab in APS.
RE: Many thanks, this has now been added to text and references
The authors have assessed the criticisms raised by the referee.
However, there are still some issue that should be addressed.
The main point is the inconsistent use of acronyms, it seems that the paper has not been checked. The reader gets lost!
- The acronym for anti-phospholipid antibodies has been inserted in line 36, and is aPL. Throughout the manuscript, one can read aPLs, aPL antibodies (line 130, 135, 159), again anti-phospholipid antibodies in extenso (e.g. line 86, 87)
RE: Many thanks, we have now modified the manuscript reporting only the acronym aPL after the first citation of it in line 36
- The acronym for anti-cardiolipin antibodies has been inserted in line 36, and is aCL. Throughout the manuscript, one can read aCL, aCA,magain anti-cardiolipin antibodies in extenso.
RE: Many thanks, we have now modified the manuscript reporting only the acronym aCL after the first citation of it in line 36
- The same for LA.
RE: We have now adopted only LA after the first citation of Lupus Anticoagulant in the text
- Please refer to low dose acetyl salycilic acid and non aspirin. Again, please be consistent with acronyms (LDA, LD-ASA).
RE: We have now adopted the term low dose acetyl salycilic acid and its acronym (LDA) and replaced it within the text
- please insert the abbreviation the first time you cite it, e.g. for HCQ.
This has now been inserted, and HCQ has been replaced the first time the term Hydroxychloroquine has been adopted
- HydroXychloroquine (line 319)
This term has now been modified,thank you
- Sometimes the text is repetitive (and even conflicting):
- e.g. line 185: please erase the sentence “According to ISTH guidelines, an aPL profile can be regarded as responsible…”
Thank you this sentence has now been deleted
- LDA primary prophylaxis: although there is no evidence from RCT, in patients with high risk profile (double/triple aPL positivity) it is common practice to prescribe primary thromboprophylaxis with LDA. Please be consistent (line 228 versus line 245)
RE: We have now modified the first sentence reporting” however this practice is not supported by evidences from randomized controlled trials”, this makes the following sentence consistent
“The protective role of LDA for the primary prophylaxis of thrombosis is, uptodate, not supported by robust evidences, also for those patients with high risk persistent aPL positivity. Primary prevention of thrombosis should be thus defined on an individual basis in patients with high risk aPL, It should be based on regular assay of antibodies titer and control of any other modifiable risk factor for thrombosis, along with treatment of any underlying known autoimmune disease [43], whether or not within this context, risk strati-fication may include also additional “non criteria” manifestations like thrombocytopenia needs to be clarified yet
- Line 273: Pregnant women with thrombotic APS should be treated. Please erase “or a history of VTE and/or arterial thrombosis related to aPL: this is thrombotic APS)
RE: Thank you, this sentence has now been deleted
- Please do not include management of carriers in Table 2, as this is debatable.
RE: Thank you, this line has now been deleted in Table 2
Line 59: Please add non criteria MANIFESTATION
RE: Thank you, the term manifestation has now been added
Line 59: Please rephrase to “This is a still gray area, indeed in the Task Force Report….”
RE: Thank you, this sentence has now been rephrased this way”: this is however still debated, ,the last reported meeting of the Technical Task Force Report on Antiphospholipid Syndrome Clinical Features, suggested in fact to consider thrombocytopenia as a classification criterion of APS [13].
Line 330: Please cite RITAPS study for rituximab in APS.
RE: Many thanks, this has now been added to text and references